# Does GNN Pretraining Help Molecular Representation?

**Ruoxi Sun**
Google Cloud AI Research
ruoxis@google.com

**Hanjun Dai**
Google Research, Brain Team
hadai@google.com

**Adams Wei Yu**
Google Research, Brain Team
adamsyuwei@google.com

## Abstract

Extracting informative representations of molecules using Graph neural networks (GNNs) is crucial in AI-driven drug discovery. Recently, the graph research community has been trying to replicate the success of self-supervised pretraining in natural language processing, with several successes claimed. However, we find the benefit brought by self-supervised pretraining on small molecular data can be negligible in many cases. We conduct thorough ablation studies on the key components of GNN pretraining, including pretraining objectives, data splitting methods, input features, pretraining dataset scales, and GNN architectures, to see how they affect the accuracy of the downstream tasks. Our first important finding is, self-supervised graph pretraining do not always have statistically significant advantages over non-pretraining methods in many settings. Secondly, although noticeable improvement can be observed with additional supervised pretraining, the improvement may diminish with richer features or more balanced data splits. Thirdly, hyper-parameters could have larger impacts on accuracy of downstream tasks than the choice of pretraining tasks, especially when the scales of downstream tasks are small. Finally, we provide our conjectures where the complexity of some pretraining methods on small molecules might be insufficient, followed by empirical evidences on different pretraining datasets.

## 1 Introduction

Graph neural networks (GNNs) , due to their effectiveness, have been adopted to model a wide range of structured data, such as social networks, road graphs, citation networks, etc. Molecule modeling is one of these important applications, where it serves as the foundation of biomedicine and nurturing techniques like novel drug discovery. However, labeling biomedical data are usually time-consuming and expensive and thus task-specific labels are extremely inadequate. This poses a big challenge to the field. Recently, inspired by the remarkable success of self-supervised pretraining from natural language processing [6, 2, 28] and computer vision domains [11, 5], researchers start trying to apply the pretrain-finetune paradigm to molecule modeling with GNN, hoping to boost the performance of various molecular tasks by pretraining the model on the enormous unlabeled data. For instance, many methods have been proposed [32, 29], where significant performance improvements are claimed by pretraining on large scale datasets [12, 22, 35–37]. Despite of the promising results, we find that reproducing some of these outstanding gains via graph pretraining can be non-trivial, and sometimes the improvement largely relies on the experimental setup and the extensive hyper-parameter tuning of downstream tasks, rather than the design of pretraining objectives. These observations motivate us to rethink the effectiveness of graph pretraining with unsupervised or self-supervised objectives, and investigate what factors would influence the effectiveness of self-supervised graph pretraining.

In this paper, we perform systematic studies to assess the performance of popular graph pretraining objectives on different types of datasets, and exploit various confounding components in experimental

36th Conference on Neural Information Processing Systems (NeurIPS 2022).

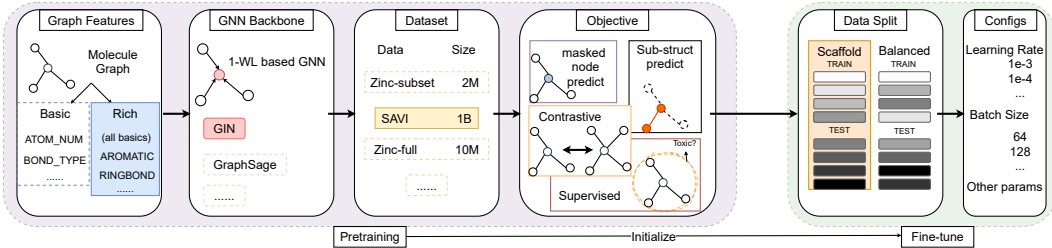

Figure 1: A typical pipeline for graph pretraining and deployment for downstream applications.

setup in deciding the performance of downstream tasks with or without pretraining. Here, we restrict our studies to small molecular graphs, as opposed to other application domains, such as social networks or citation graphs. The key insights and take-aways of this paper are:

- Among the pretraining tasks we evaluated, the self-supervised pretraining alone does not provide statistically significant improvements over non-pretrained methods on downstream tasks.

- When additional supervised pretraining step is conducted after self-supervised pretraining, we observe statistically significant improvements. However, the gain becomes marginal on some specific data splits or diminishes if richer features are introduced.

- Beyond data splits and hand-crafted features, the usefulness of graph pretraining is also sensitive to the experimental hyperparameters, such as learning rates and number of study repeats. Different setups can lead to opposite conclusions.

- In conclusion, different from the previous works, we do not observe clear and unconditional gains achieved by graph pretraining on molecular representation, indicating it is still too early to conclude graph pretraining is effective in molecular domain.

- We investigate the reason of above and hypothesize that the complexity of some pretraining methods on molecules is insufficient, leading to less transferable knowledge for downstream tasks.

Despite the overall negative results we obtained, the main goal of this paper is not to discourage the pretraining research for small molecules. Instead, we hope to raise the attention on different aspects of experiments and the role of simple hand-crafted features, so as to provide useful information for designing better pretraining approaches. Below we first introduce the background of GNN and its pretraining in Section 2, and then our experimental design and results in Section 3 and Section 4, respectively. Finally we conclude with our findings and the limitations in Section 5 and Section 6.

## 2 Preliminary

Table 1: Summary of Experiments. Table 12 and Table 13 are deferred to appendix due to space limit.

| | Pretrain Objective | | Graph Features | | Downstream Splits | | GNN Arch | | Pretrain Dataset | |
|---|---|---|---|---|---|---|---|---|---|---|
| | Self-Supervised | Supervised | Rich | Basic | Balanced | Scaffold | GIN | GraphSage | ZINC15 | SAVI |
| Table 2 | ✔ | | ✔ | | ✔ | | ✔ | | ✔ | |
| Table 3 | | ✔ | ✔ | | ✔ | | ✔ | | ✔ | |
| Table 4 | ✔ | | ✔ | | | ✔ | ✔ | | ✔ | |
| Table 5 | | ✔ | ✔ | | | ✔ | ✔ | | ✔ | |
| Table 6 | ✔ | | | ✔ | ✔ | | ✔ | | ✔ | |
| Table 7 | | ✔ | | ✔ | ✔ | | ✔ | | ✔ | |
| Table 8 | ✔ | | | ✔ | | ✔ | ✔ | | ✔ | |
| Table 9 | | ✔ | | ✔ | | ✔ | ✔ | | ✔ | |
| Table 10 | ✔ | ✔ | ✔ | | ✔ | | ✔ | | | ✔ |
| Table 11 | ✔ | ✔ | ✔ | | | ✔ | ✔ | | | ✔ |
| Table 12 | ✔ | ✔ | ✔ | | ✔ | | | ✔ | ✔ | |
| Table 13 | ✔ | ✔ | ✔ | | | ✔ | | ✔ | ✔ | |

**Graph Neural Networks (GNNs).** Let $G = \{V, E\}$ denote a molecule graph with $V$ as the set of nodes and $E$ as the set of edges. Given the node features $X_i$, most GNNs learn an embedding representation $h_i$ for every node $i \in V$ by aggregating representations from connected nodes and

edges, denoted as graph convolution. These procedure repeats for $K$ times with the update equation as follows:

$$h_i^k = \text{UPDATE}(h_i^{k-1}, \text{AGGREGATE}(\{h_i^{k-1}, h_j^{k-1}, e_{ij}\} : \forall j \in N(i)))$$

where $\mathcal{N}(i)$ is the set of neighbor nodes of $i$ and $h_i^0 = X_i$. The representation for entire graph $G$ is then obtained by permutation-invariant transformation on node representation, $h_G = \text{READOUT}(h_i^K | i \in V)$. In this paper we mainly study the GNNs that belong to this family, namely the WL-1 GNNs.

**Finetune.** After the pretraining, the pretrained model is used to finetune on the downstream tasks. For molecule property prediction tasks, the graph-level representation obtained from the pretrained model is connected to linear classifiers to predict downstream task labels. The fine-tuning is performed in an end-to-end manner, where both the pretrained GNN and the linear classifiers are trainable.

**Graph pretraining objectives.** The primary goal of pretraining is to learn representations with robust transferable knowledge of graphs from the abundant pretraining data and then generalize to downstream tasks with usually different supervision signals. Generally the pretraining objectives can be categorized into self-supervised and supervised ones. We present a brief overview of some representative objectives in the following sections.

## 2.1 Self-supervised (unsupervised) pretraining

In self-supervised pretraining, the pretraining objective is designed to learn self-generated targets from the structure of the molecules, such as the type of nodes and edges, prediction of local context, graph partition, node clustering, occurrence of some functional groups, and etc. The predictive target can be node/edge level or entire graph level. We present some representative ones below:

### 2.1.1 Node Prediction

Node prediction is a node-level classification task given the masked context of entire graph. Similar to Devlin et al. [6], some portion of node attributes are masked and replaced with mask-specified indicators in the node input feature. After graph convolution, the embedding output from GNN is used to predict the true attribute of the node, e.g. atom type in molecular graphs, through a linear classifier on top of the node embedding.

### 2.1.2 Context Prediction

Context prediction task is a sub-graph level task aiming at learning embedding that can represent the local subgraph surrounding a node. Generally it can be viewed as a masked task for substructure. Since it is essentially a structured prediction which can be difficult in general, Hu et al. [12] leverages the adversarial learning to teach the model to distinguish the positive sub-graph embedding from the negative ones. Rong et al. [22] instead builds a dictionary of structures that captures the property of sub-graphs (e.g. type and quantity of neighbour nodes and bonds), and turns it into a multi-class classification problem.

### 2.1.3 Motif Prediction

Motif prediction [22] is to predict the existence of functional groups, such as benzene ring or hydroxyl. The motifs are extracted automatically from RDKit [15] . The motif prediction task is formulated as a graph-level multi-label binary classification task, where the graph embedding is used to jointly predict the occurrence of these semantic functional motifs.

### 2.1.4 Contrastive learning

Graph contrastive learning is to maximize the agreement of two augmented views of the same graph, and minimize the agreement of different graphs. The optimization is conducted using contrastive loss in the latent embedding space [35, 10, 37, 29]. The augmentation function needs to transform graphs into realistic and novel augmentations without affecting semantic labels of the graphs. For example, the transformation can be small perturbations or modifications on node/edge embedding, drop of a

Table 2: **Self-supervised** + Rich feature + Balanced Scaffold Split. No pretrain has an average value of 78.0% over all 5 datasets.

| Methods | BBBP | BACE | TOX21 | TOXCAST | SIDER | AVE GAIN |
|---|---|---|---|---|---|---|
| **No pretrain** | 92.23($\pm$3.07) | 87.43($\pm$1.63) | 79.20($\pm$1.99) | 69.13($\pm$0.55) | 61.92($\pm$0.89) | 0($\pm$1.626) |
| **Node Prediction** | 92.24($\pm$2.76) | 87.32($\pm$1.67) | 79.57($\pm$2.03) | 69.77($\pm$0.13) | 61.62($\pm$1.12) | 0.122($\pm$1.542) |
| **Context Prediction** | 92.68($\pm$1.19) | 86.98($\pm$1.26) | 79.05($\pm$2.51) | 70.18($\pm$0.44) | 61.65($\pm$0.77) | 0.126($\pm$1.234) |
| **Motif Prediction** | 92.63($\pm$1.19) | 87.16($\pm$1.66) | 79.22($\pm$2.38) | 69.09($\pm$0.07) | 62.45($\pm$1.25) | 0.128($\pm$1.310) |
| **Contrastive learning** | 92.31($\pm$1.58) | 86.67($\pm$2.40) | 78.45($\pm$2.44) | 68.37($\pm$0.80) | 61.22($\pm$1.20) | -0.578($\pm$1.684) |

few nodes or edges, and so on. These transformations enforce an underlying prior for contrastive learning, that is, local transformation does not change the semantic meaning of a graph.

## 2.2 Supervised pretraining

Supervised pretraining aims to learn domain-specific graph-level knowledge from specifically designed pretraining tasks. For molecular application, the supervised labels are generated from a diverse set of functional studies like biochemical assays. The pretraining task is to perform multiple binary classification and jointly learn the supervised labels. Although the pretraining mainly refers to unsupervised or self-supervised methods as they are not limited by the requirement of supervised labels, supervised pretraining is still a great source to investigate the graph pretraining in general.

Table 3: **Supervised** + Rich feature + Balanced Scaffold. No pretrain has an average AUC of 78.0%.

| Methods | BBBP | BACE | TOX21 | TOXCAST | SIDER | AVE GAIN |
|---|---|---|---|---|---|---|
| **No pretrain** | 92.23($\pm$3.07) | 87.43($\pm$1.63) | 79.2($\pm$1.99) | 69.13($\pm$0.55) | 61.92($\pm$0.89) | 0($\pm$1.626) |
| **Supervised** | 91.65($\pm$2.11) | 86.91($\pm$1.86) | 81.13($\pm$2.39) | 71.64($\pm$0.46) | 62.14($\pm$1.13) | 0.712($\pm$1.590) |
| **Masking Node + Supervised** | **93.43($\pm$2.50)** | 86.90($\pm$2.04) | **81.93($\pm$1.79)** | 71.66($\pm$0.73) | 62.68($\pm$1.82) | 1.338($\pm$1.776) |
| **Context Prediction + Supervised** | 92.27($\pm$1.57) | **88.72($\pm$1.68)** | 81.71($\pm$1.79) | **72.19($\pm$0.79)** | **63.21($\pm$1.49)** | 1.638($\pm$1.464) |

# 3 Experiment framework

To investigate pretraining on graphs for molecule representations, we first revisit the typical pretraining-finetuning pipeline used in the literature. Figure 1 shows the overall procedure of deployment, with several design choices presented at each stage of the pipeline. Since different choices at each stage can lead to different performances on the downstream tasks, we investigate them one at a time while keeping others the unchanged. The design principle of our experiment framework is to analyze the effect of *every* stage in the pipeline as comprehensive as possible, while also keeping it tractable to avoid exponentially many experiments.

## 3.1 Design choices

We consider the design choices for the four pretraining objectives.

**Pretraining objective**   In Section 2 we have provided a brief literature review over the pretraining methods for molecule representation. Here we categorize those pretraining by different principles, and present one well-recognized representative of each category. The representatives are selected because they have more desired properties, such as better performance, compared with their counterparts.

- **Masking.** We leverage the node prediction objective, which randomly masks 15% of the nodes' feature and then ask GNN to make prediction on the node attributes of the masked ones. This strategy resembles the BERT pretraining [6] in natural language processing.

- **Structured.** Unlike text data where the topology is a sequence, the graph has rich structure information. Following Hu et al. [12], we use context prediction objective, which masks out the context from $k_1$-hops to $k_2$-hops and leverages adversarial training to predict the true context embeddings from the random context embeddings.

- **Graph-level self-supervised.** Following [22], GNN is asked to predict whether a motif is contained in a molecule. The motif can be extracted from the molecule with RDKit [15]. The motifs are 85 motifs [1] for multi-label classification.

---

[1] http://rdkit.org/docs/source/rdkit.Chem.Fragments.html

Table 4: Self-supervised + Rich feature + **Scaffold**. No pretrain has an average ROC-AUC of 71.8% over all benckmark datasets.

| Methods | BBBP | BACE | TOX21 | TOXCAST | SIDER | AVE GAIN |
|---|---|---|---|---|---|---|
| No pretrain | 74.83($\pm$0.73) | 80.10($\pm$0.42) | 75.86($\pm$0.58) | 65.95($\pm$0.15) | 62.30($\pm$1.14) | 0($\pm$0.579) |
| Node Prediction | 73.45($\pm$0.27) | 83.66($\pm$0.75) | 75.30($\pm$0.37) | 66.50($\pm$0.06) | 65.08($\pm$0.12) | 0.990($\pm$0.323) |
| Context Prediction | 74.10($\pm$0.22) | 81.87($\pm$0.49) | 75.37($\pm$0.11) | 66.86($\pm$0.07) | 62.84($\pm$0.46) | 0.400($\pm$0.280) |
| Motif Prediction | 73.65($\pm$0.36) | 80.58($\pm$2.04) | 74.55($\pm$0.79) | 65.63($\pm$0.07) | 64.05($\pm$0.23) | -0.116($\pm$0.766) |
| Contrastive learning | 73.32($\pm$2.38) | 80.51($\pm$0.80) | 74.55($\pm$0.22) | 65.70($\pm$0.09) | 64.39($\pm$0.63) | -0.114($\pm$0.513) |

Table 5: Supervised + Self-supervised + Rich feature + **Scaffold**. No pretrain get 71.8% average ROC-AUC.

| Methods | BBBP | BACE | TOX21 | TOXCAST | SIDER | AVE GAIN |
|---|---|---|---|---|---|---|
| No pretrain | **74.83**($\pm$**0.73**) | 80.10($\pm$0.42) | 75.86($\pm$0.58) | 65.95($\pm$0.15) | 62.30($\pm$1.14) | 0($\pm$0.604) |
| Supervised | 72.79($\pm$0.7) | 83.23($\pm$0.67) | 77.66($\pm$0.08) | 67.72($\pm$0.13) | 65.34($\pm$0.17) | 1.540($\pm$0.350) |
| Masking Node + Supervised | 73.38($\pm$0.55) | **84.42**($\pm$**0.27**) | **77.85**($\pm$**0.24**) | 67.14($\pm$0.28) | 64.06($\pm$0.28) | 1.562($\pm$0.324) |
| Context Prediction + Supervised | 73.81($\pm$0.52) | 84.35($\pm$0.93) | 77.11($\pm$0.14) | 67.87($\pm$0.08) | **65.19**($\pm$**0.17**) | 1.858($\pm$0.368) |

- **Contrastive.** We generate two views of the same graph by corrupting the input node features with Gaussian noise. We leverage the contrastive learning loss proposed in [35]: we maximize the consistency between positive pairs (from same graphs) and minimize that between negative pairs (from different graphs). In this paper, we restrict ourselves to this specific contrastive training method, however, various contrastive learning methods can be further explored.

- **Graph-level supervised.** Finally when applicable, we use the ChEMBL dataset with graph-level labels for graph-level supervised pretraining as Hu et al. [12].

The above are the design choices for pretraining objectives. Next, we consider other factors that influence graph-pretraining performance.

**Graph Features**   Each molecule is represented by a graph with atoms as nodes and bonds as edges. In this paper we mainly consider the graph representations without the 3D information. For each molecule graph, chemical properties of nodes and edges are extracted to serve as node and edge features for the graph neural networks. Depending on how rich the features are, we categorize the design choices into two categories:

- **Basic features.** The basic set of features are the ones used in Hu et al. [12]. Specifically, the node features contain the atom type and the derived features, such as formal charge list, chirality list, etc. The edge features contain the bond types and the bond directions. These features are categorical, and thus will be encoded in a one-hot vector individually and then concatenated together to form the feature vector for node/edge representation.

- **Rich features.** The rich feature set is a superset of the basic features. In addition to the basic ones mentioned above, it comes with the additional node features such as hydrogen acceptor match, acidic match and bond features such as ring information. This set of features are used in Rong et al. [22]. Additionally and importantly, we follow their setting to incorporate additional 2d normalized rdNormalizedDescriptors features [2], which is used in the downstream tasks only and not in pretraining.

Please refer to the original papers for the full set of basic [12] and rich [22] features, respectively.

**GNN Backbone**   The GNN architecture also plays a role in graph pretraining. In Hu et al. [12], the results show that pretraining on GNN variants like GIN [33] would improve the performance on downstream tasks, while the performance with architectures like GAT [27] would actually get worse performance with pretraining. As the GNNs based on 1-Weisfeiler-Lehman (WL) test have similar representation power [33] bounded by the Weisfeiler-Lehman isomorphism check [23], we consider the two representative GNN architectures, namely the **GIN** [33] and **GraphSage** [9]. They have shown benefits with graph pretraining in Hu et al. [12].

**Pretraining dataset**   In natural language pretraining, researchers observed a significant performance boost due to self-supervised pretraining on large-scale data, that is, the larger the pretraining dataset

---

[2]`https://github.com/bp-kelley/descriptastorus` for the feature descriptor.

Table 6: Self-supervised + **Basic feature** + Balanced Scaffold. No pretrain has an average AUC of 76.7% over all 5 datasets.

| Methods | BBBP | BACE | TOX21 | TOXCAST | SIDER | AVE GAIN |
|---|---|---|---|---|---|---|
| **No pretrain** | 91.46($\pm$ 0.85) | 84.29($\pm$ 3.80) | 78.35($\pm$ 0.95) | 68.31($\pm$ 1.61) | 61.15($\pm$ 2.46) | 0($\pm$1.934) |
| **Node Prediction** | 91.23($\pm$ 1.51) | 84.97($\pm$ 1.55) | 77.77($\pm$ 1.23) | 68.98($\pm$ 1.11) | 61.20($\pm$ 0.41) | 0.118($\pm$1.162) |
| **Context Prediction** | 92.13($\pm$ 1.04) | 84.83($\pm$ 3.19) | 78.79($\pm$ 2.52) | 68.29($\pm$ 1.23) | 62.32($\pm$ 2.99) | 0.560($\pm$2.194) |

Table 7: Supervised + Self-supervised + **Basic feature** + Balanced Scaffold. No pretrain has an average AUC of 76.7%.

| Methods | BBBP | BACE | TOX21 | TOXCAST | SIDER | AVE GAIN |
|---|---|---|---|---|---|---|
| **No pretrain** | 91.46($\pm$ 0.85) | 84.29($\pm$ 3.80) | 78.35($\pm$ 0.95) | 68.31($\pm$ 1.61) | 61.15($\pm$ 2.46) | 0($\pm$ 1.934) |
| **Supervised** | 90.70($\pm$ 0.74) | 84.22($\pm$ 2.69) | 80.45($\pm$ 1.47) | 69.47($\pm$ 1.06) | **63.38($\pm$ 1.44)** | 0.932($\pm$ 1.480) |
| **Masking Node + Supervised** | 91.10($\pm$ 2.88) | 85.54($\pm$ 4.57) | **81.49($\pm$ 1.52)** | 70.77($\pm$ 1.00) | 62.81($\pm$ 2.61) | 1.630($\pm$ 2.516) |
| **Context Prediction + Supervised** | **91.54($\pm$ 3.52)** | **85.71($\pm$ 2.92)** | 81.23($\pm$ 1.94) | **71.36($\pm$ 1.05)** | 62.75($\pm$ 2.27) | 1.806($\pm$ 2.340) |

is, the better the downstream performance it is [20]. Inspired by this success in natural language processing, we test the algorithms on two unlabeled pretraining datasets with different scales.

- **ZINC15 [25]**: ZINC15 contains 2 million molecules. This dataset was preprocessed following Hu et al. [12].
- **SAVI [19]**: The SAVI dataset contains about 1 billion molecules, which are significantly larger than ZINC15. To the best of our knowledge, it has never been used for pretraining tasks before. This dataset contains drug-like molecules synthesized by computer simulated reactions.

Additionaly, we used **ChEMBL [8]** as the supervised datasets. Different from the above ZINC15 and SAVI dataset which are only used for self-supervised pretraining, this dataset contains 500k drug-able molecules with 1,310 prediction target labels from bio-activity assays for drug discovery. Thus like in Hu et al. [12] we only leverage it for *supervised* pretraining.

**Data split on downstream tasks**    The downstream tasks for molecular domain we used are 5 benchmark datasets from MoleculeNet [30] (See Appendix A.5 for more details). The train/valid/test sets are split with ratio 8:1:1. For molecule domain, the random split is not the most meaningful way to assess the performance, because the real-world scenarios often require generalization ability on out-of-distribution samples. So we consider the following ways to split the data:

- **Scaffold Split [12, 21]** This strategy first sorts the molecules according to the scaffold (e.g. molecule structure), and then partition the sorted list into train/valid/test splits consecutively. Therefore, the molecules in train and test sets are most different ones according to their molecule structure. Note this strategy would yield deterministic data splits.
- **Balanced Scaffold Split [1, 22]** This strategy introduces the randomness in the sorting and splitting stages above, thus one can run on splits with different random seeds and report the average performance to lower the evaluation variance.

We choose balanced scaffold as our major evaluation configuration, because it allows us to evaluate the algorithm on multiple data splits while maintaining the ability to evaluate out of distribution samples (e.g. assess generalization ability). Evaluating on one single split (such as scaffold split) can be subject to bias due to one specific split, leading to higher variance in evaluation.

## 3.2   Experiment protocol

As the total number of configurations for the entire pipeline can be combinatorially large which is not practical for us to exhaustively experiment with all of them, we design our protocol with a pairwise comparison principle. Specifically, we first anchor a *vanilla configuration* with a certain design choice of combination for each stage. To study the effect of each stage on the pretraining effectiveness, we vary the design choice one stage at a time compared to the *vanilla configuration*.

For all these experiments, to assess the effectiveness of graph pretraining, we report the **ROC-AUC** on downstream tasks as well as the relative **average gain** over all downstream datasets with and without pretraining. For each setting we will report the mean and standard deviation (in parenthesis) over three runs with different random seeds. We tune the model on downstream tasks with the validation set, and report the evaluation metric on the test set using the model with best validation

Table 8: Unsupervised + **Basic feature + Scaffold**. No pretrain has an average accuracy of 68.7% over all benckmark datasets.

| Methods | BBBP | BACE | TOX21 | TOXCAST | SIDER | AVE GAIN |
|---|---|---|---|---|---|---|
| **No pretrain** | 69.62($\pm$ 1.05) | 75.77($\pm$4.29) | 75.52($\pm$0.67) | 63.67($\pm$0.32) | 59.07($\pm$1.13) | 0($\pm$1.492) |
| **Node Prediction** | 68.70($\pm$2.16) | 76.95($\pm$0.12) | 75.88($\pm$0.60) | 64.11($\pm$0.38) | 61.29($\pm$0.87) | 0.656($\pm$0.826) |
| **Context Prediction** | 69.41($\pm$1.44) | **81.96**($\pm$0.72) | 75.49($\pm$0.75) | 63.48($\pm$0.31) | **62.27**($\pm$**0.90**) | 1.792($\pm$0.824) |

Table 9: Supervised + **Basic feature + Scaffold**. No pretrain has an average accuracy of 68.7% over all benckmark datasets.

| Methods | BBBP | BACE | TOX21 | TOXCAST | SIDER | AVE GAIN |
|---|---|---|---|---|---|---|
| **No pretrain** | 69.62($\pm$ 1.05) | 75.77($\pm$4.29) | 75.52($\pm$0.67) | 63.67($\pm$0.32) | 59.07($\pm$1.13) | 0($\pm$1.492) |
| **Supervised** | 68.96($\pm$0.64) | 76.30($\pm$1.30) | 76.64($\pm$0.39) | 66.07($\pm$0.22) | 61.97($\pm$0.96) | 1.258($\pm$0.702) |
| **Masking Node + Supervised** | **71.41**($\pm$**0.67**) | 84.59($\pm$0.35) | **79.13**($\pm$**0.29**) | 65.32($\pm$0.37) | 62.12($\pm$0.19) | 3.784($\pm$0.374) |
| **Context Prediction + Supervised** | 69.63($\pm$0.25) | **83.34**($\pm$**0.67**) | 78.11($\pm$0.28) | **66.15**($\pm$**0.48**) | **63.48**($\pm$**0.43**) | 3.412($\pm$0.422) |

performance. For each setup, we report the average performance obtained with three random seeds. We tune the learning rate in $\{1e^{-4}, 5e^{-4}, 1e^{-3}, 5e^{-3}, 1e^{-2}, 5e^{-2}, 1e^{-1}\}$ for each setup *individually* and select the one with best validation performance. For GNNs we fix the hidden dimension to 300 and number of layers to 5.

## 4 Results

In this section, we present the results and discussions for a set of experiments designed with the protocols in Section 3.2. Table 1 summarizes the experimental configurations for each following table. We will elaborate on them in the following sections. Due to space limit, we defer our investigation on different GNN architectures to appendix (Section A.1).

### 4.1 Vanilla configuration

We choose the *vanilla configuration* with the settings from existing works [12, 22]. Specifically, we use the **rich feature** with **GIN** backbone, pretrained on **ZINC15** when pretraining is applied, and evaluate on the **Balanced Scaffold Split** for downstream tasks. One important baseline is without pretraining. For the ease of comparison, we include the results without pretraining in each table.

### 4.2 Self-supervised pretraining objectives

We compare the results pretrained with different self-supervised pretraining objectives. As is presented in Section 3.1, we consider four representative types of pretraining objectives. For the ease of comparing the performance, we only consider one objective at a time, instead of mixing different pretraining objectives to obtain a multi-task pretrained model. Table 2 shows the performance on downstream molecule property prediction benchmarks with models initialized from different pretraining objectives. The relative average gain compared to the one without pretraining is not statistically significant, i.e., not larger than the standard deviations of multiple runs. All the four different objectives obtain similar gains/loses regardless of very different designs. To fully understand the effect of self-supervised pretraining on molecule representation, we further investigate the performance of different pretraining objectives in combination with other factors, such as input features or data splits, as described in the following sections.

### 4.3 Supervised pretraining objectives

In addition to the self-supervised objectives, we study the potential benefits with supervised pretraining. Unlike the self-supervised setting where the molecule graphs themselves are used for pretraining, the supervised pretraining requires extra cost of data labeling, and thus is not scalable for large scale pretraining. In this paper, we present the results with supervised pretraining alone, as well as the joint pretraining. e.g. pretrain with self-supervised objective and followed by supervised pretraining, in Table 3. We can see with the supervised pretraining, one can improve the downstream performance, which aligns with the observation from Hu et al. [12]. Our hypothesis is that, supervised pretraining is helpful when the pretraining tasks are closely aligned with the downstream tasks. In particular,

Table 10: **Large scale pretraining data** with balanced scaffold split. No pretraining gets an average AUC of 78.0%.

| Methods | BBBP | BACE | TOX21 | TOXCAST | SIDER | AVE GAIN |
|---|---|---|---|---|---|---|
| **No pretrain** | 92.23(±3.07) | 87.43(±1.63) | 79.2(±1.99) | 69.13(±0.55) | 61.92(±0.89) | 0(±1.626) |
| **Node Prediction** | 92.33(±2.08) | 87.22(±1.79) | 79.12(±1.62) | 69.47(±0.65) | 61.24(±1.94) | -0.106(±1.616) |
| **Context Prediction** | 93.32(±0.53) | 87.77(±2.94) | 79.18(±2.48) | 70.13(±0.56) | 62.24(±2.65) | 0.546(±1.832) |
| **Masking Node + Supervised** | 93.23(±3.02) | 86.39(±1.67) | 81.89(±1.58) | 71.77(±0.50) | 63.73(±2.20) | 1.420(±1.794) |
| **Context Prediction + Supervised** | 92.55(±2.93) | 87.76(±1.87) | 82.19(±1.58) | 72.91(±0.71) | 62.44(±0.45) | 1.588(±1.508) |

the bio-activity labels provided by ChEMBL is highly related to the drug discovery purpose and drug discovery properties are the major topics evaluated in the downstream tasks. Therefore, the positive correlation between the pretraining supervision and downstream tasks contribute the most to the performance improvement of downstream tasks.

## 4.4 Data split on downstream tasks

Molecular data is usually diverse and limited, so chemists are particularly interested in the generalization ability of GNNs on out of distribution data. Also due to the same reason (i.e. limited and diverse data), the variance in performance of different splits is significant, which poses challenges on robust evaluation. In *vanilla configuration* we use the balanced scaffold split, and here we show additional results with the **scaffold** split, which is a deterministic data split that makes the train/valid/test set differ from each other the most. Table 4 and Table 5 respectively present the results using scaffold split with self-supervised without and with additional supervised pretraining. Compared with Table 2 and Table 3, it is clear to see that Table 4 and Table 5 have significantly lower ROC-AUC. Specifically the AUC drops 6.2% on average for all benchmarks without pretraining. On the other hand, we can see if we compare Table 4 with Table 2, or Table 5 with Table 3 respectively, the gain of pretraining is more significant on the **scaffold** split. We speculate the reason for the improvement of scaffold split is that the initialization of neural network parameters (e.g. from pretraining) are typically critical for the out-of-distribution generalization (e.g. scaffold split). Similar observations have also been studied in the meta-learning literature [7]. Although the gain with supervised pretraining is significant in Table 5, the effect of self-supervised pretraining is mixed in Table 4. This indicates the effectiveness of self-supervised pretraining on scaffold split is not significant enough to claim "very helpful".

## 4.5 Graph features

So far we have presented the results with rich features. Now we want to see how those basic features used in Hu et al. [12] affect the outcome. Table 6 and Table 7 show the test ROC-AUC (%) performance with *basic features* on the balanced scaffold splits using self-supervised or supervised pretraining objectives, respectively. Table 8 and Table 9 show the same results but on scaffold split.

In a nutshell, without pretraining, rich features lead to an average gain of 1.3% and 3.1% over basic features using balanced scaffold split and scaffold split, respectively. Specifically, it achieves 76.7% vs 78.0% for balanced scaffold split, and 68.7% vs 71.8% on scaffold split. The gain brought by the rich features are more significant than the ones with different self-supervised pretraining objectives. Table 6 to Table 9 show that pretraining has more positive impact when basic features are used. In particular, the self-pretraining with context prediction shows significant gains especially in the scaffold split setting. However, the gain diminishes when careful feature engineering are applied to the downstream tasks (use rich feature in *vanilla configuration*). The supervised pretraining continues the significant gain under these settings, which shows the consistency and reliability of the situation with the labeled and downstream-task-aligned supervisions.

## 4.6 Pretraining datasets

As observed in natural language processing domain, more text pretraining data lead to better downstream performance. Intuitively this can be true for molecule representation domain as well, so we run a new set of experiments with the model pretrained on SAVI dataset, which is about 500 times larger than the ZINC15 dataset we used in the above result sections. We present the results pretrained on SAVI dataset using balanced scaffold split or scaffold split in Table 10 and Table 11, respectively. Other configurations are the same as the *vanilla configuration*.

Table 11: **Large scale pretraining** data with scaffold split. No pretraining gets an average AUC of 71.8%.

| Methods | BBBP | BACE | TOX21 | TOXCAST | SIDER | AVE GAIN |
|---|---|---|---|---|---|---|
| No pretrain | 74.83(±0.73) | 80.10(±0.42) | 75.86(±0.58) | 65.95(±0.15) | 62.30(±1.14) | 0(±0.604) |
| Node Prediction | 73.81(±1.82) | 81.90(±1.59) | 74.94(±0.05) | 66.95(±0.12) | 62.93(±0.34) | 0.298(±0.784) |
| Context Prediction | 74.32(±0.85) | 83.93(±0.24) | 74.42(±0.19) | 67.01(±0.29) | 64.83(±0.45) | 1.094(±0.404) |
| Masking Node + Supervised | 73.32(±0.60) | 83.38(±1.05) | 78.59(±0.09) | 67.01(±0.18) | 65.40(±0.12) | 1.732(±0.408) |
| Context Prediction + Supervised | 74.38(±0.93) | 86.33(±0.16) | 78.16(±0.25) | 68.71(±0.07) | 62.22(±0.48) | 2.152(±0.378) |

Compared with the performance on ZINC15, the SAVI pretraining data does not lead to a significant improvement either on balanced scaffold split (Table 2 vs Table 10) or scaffold split (Table 4 or Table 11). Similarly, the self-supervised pretraining objectives lead to negligible gain on downstream task performance, while the supervised one still achieves a clear gain.

As the result is counterintuitive, we further investigate the reason behind it by inspecting the pretraining performances with different training objectives on both ZINC15 and SAVI datasets. We plot the curve of accuracy growth with the number of training steps iterated. We can see from Figure 2 that in all settings the pretraining accuracy grows above 90% quickly after only 0.1 to 0.2M steps and also converges quickly. Given that the model gets very high accuracy without even going through 1 epoch of the SAVI dataset, it is expected that the larger training data like SAVI may not provide more learning signals for the model, and partially explains why more molecules wouldn't help significantly in this case. Furthermore, these figures might suggest several reasons of why the self-supervised pretraining may not be very effective in some situations:

- **Tasks are easy.** Some of the self-pretraining tasks for molecules might be easy, so that model learns less useful information from pretraining. For example in the masked node prediction case, the model is expected to predict the atomic number from a vocabulary with less than 100 candidate atoms. Furthermore, due to the valence constraints, the graph topology may already exclude most of the wrong atoms. As a comparison, the vocabulary size for text pretraining may be 100k or even higher. Some structured prediction tasks like context prediction might be hard, but due to the difficulty of structured prediction itself and the proposal for high quality negative examples for contrastive learning, it can still be challenging for downstream task improvements. Other strategies like motif prediction can be achieved by subgraph matching, which can be easy for GNN that intrinsically does the graph isomorphism test.

- **Data lacks diversity.** Due to biophysical and functional requirements, molecules share many common sub-structures, e.g., functional motifs. Hence, molecules may not be as diversified as text data. This is why the model learns to generalize quickly within the training distribution.

- **2D Structure is not enough to infer functionality.** Some important biophysical properties (such as 3D structure, chirality) are barely reflected in the 2D-feature-based pretraining (e.g., using smiles or 2D graph features). For example, the molecules with the same chemical formula and 2D feature, can have very different chirality, which leads to quite different toxicity [24] (e.g. flipped toxicity labels). This is not captured in the current GNN pretraining frameworks that we considered.

## 4.7 Hyper-parameters for downstream tasks

We also find that hyper-parameters for downstream tasks are critical for the their performance that their choices may change the conclusion of the effectiveness of pretraining in some settings. We can take the learning rate as an example. As the models initialized from scratch and pretraining may have different scales, the most suitable learning rate required for downstream tasks may also be different. Without tuning learning rate extensively, we may reach a misleading conclusion. In particular,

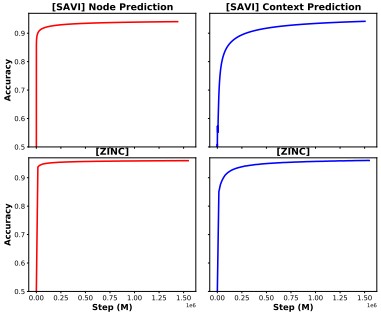

Figure 2: Pretraining accuracy on ZINC15 or SAVI datasets with node prediction or context prediction objectives.

when we adopt the default learning rate for reproducing the existing success of pretraining in Table 17 of Appendix A.4, we indeed observe the advantage of pretraining. However, if we follow our procedures (e.g. extensive search learning rate and averaging over three splits), the resulting Table 2 and Table 3 indicate no performance gain by pretraining. So we suggest that the evaluation of pretraining should consider the hyper-parameter tuning and averaging over different splits.

## 5 Summary and takeaways

Based on our experiments in Section 4, we present our takeaways by empirically summarizing our conjectures on when the pretraining would/would not help the molecular representation learning.

**When pretraining might help** We find it typically helps 1) if we can have the supervised pretraining with target labels that are aligned with the downstream tasks. However, getting large amount of high-quality and relevant supervision is not always feasible; 2) if the high quality hand-crafted features are absent. However, it seems that the gain obtained by self-supervised pretraining is not as significant as these high quality hand-crafted features based on our current studies; 3) if the downstream train, valid and test dataset distributions are substantially different.

**When the gain diminishes?** In some situations the gain of pretraining might diminish 1) if we already have the high quality hand-crafted features (e.g. rich features described in Section 3.1); 2) if we don't have the highly relevant supervisions. As shown in Section 4.6, many self-supervised pretraining tasks might be too easy for the model to learn meaningful embedding; 3) if the downstream data splits are balanced; 4) if the self-supervised learning dataset lacks diversity, despite its scale.

**Why pretraining may not help in some cases?** In our paper we pretrained a GNN on a much larger dataset (SAVI) than before, hoping to replicate gain of pretraining like in NLP domain. However, we do not obtain the expected gain. The pretraining accuracy curve (Figure 2) provides some potential explanations of why pretraining may not work: some of the pretraining accuracy curve grows above 95%+ quickly and converges fast, unlike pretraining in NLP which keeps growing to 70% and hardly plateaus. This suggests that some of the pretraining methods like masked node label prediction might be easy (as the vocabulary size is much smaller compared to NLP) and therefore transfer less knowledge for downstream tasks.

## 6 Limitations of current study

Although we have tried our best to design a comprehensive study on the effectiveness of graph pretraining for molecular representation, there are still limitations we want to point out. Due to the limited time and resources we have, we are not able to fully cover the whole picture of the current pretraining paradigm in graph neural networks. Nevertheless, we list them here in hope of preventing the over-generalization of our conclusion.

- *Distribution of graphs*. Our study focuses on pretraining for small molecule graph inductive representation learning. Recently there are works on pretraining transductive representation learning [36] on large graphs [13], where our conclusion may not be directly extended to these cases.
- *Graph architectures*. GNN is a popular research field where many new architectures with probable expressiveness are/will be proposed. The results we have shown are on two representative 1-WL GNNs. It can be possible that the latest advances of deep GNN [17] and Transformer-based GNN [34, 3, 14] might yield different results.
- *Learning objectives*. Although we have presented results with different types of self-supervised losses, there are still many variants of each type that we did not explore, like different variants [26, 31] of contrastive learning. Also, multi-task learning of different self-supervised objectives might be another direction for further exploration.
- *Downstream datasets*. We obtained our conclusion mainly on the datasets from MoleculeNet [30]. Datasets like Alchemy [4] and drug-target Interaction [18] may show different results.

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
