# A Appendix

## A.1 GNN architectures

We show results with GraphSage [9] architecture as GNN backbone in Table 12 and Table 13. We investigated the two different splits used in previous sections, as well as different self-supervised and supervised pretraining objectives. The overall performance using GraphSage architecture is comparable with results obtained using GIN architecture, and the conclusion about pretraining objectives is the same with what we obtained on the GIN as well. As generally these architectures have similar representation power [33], this outcome should be expected. Additionally, we also explored graph pretraining with graph transformer proposed in [22], which is supposed to be more expressive. However, Table 18 in Appendix shows that the results are not competitive. For higher-order GNNs or the deeper GNNs the conclusion might be different, but in general we hold a conservative view towards whether the graph architecture can make a big difference in deciding whether graph pretraining is helpful.

Table 12: **GraphSAGE** GNN architecture on Balanced Scaffold Split. The average ROC-AUC without pretraining on all benchmark datasets is 78.1%. Gray- and transparent- shaded show supervised and unsupervised pretraining objectives.

|  | BBBP | BACE | TOX21 | TOXCAST | SIDER | AVE GAIN |
|---|---|---|---|---|---|---|
| **No pretrain** | 92.46(±2.91) | 86.46(±1.63) | 78.99(±1.57) | 70.35(±0.09) | 62.18(±0.77) | 0(±1.394) |
| **Node Prediction** | 93.76(±1.70) | 88.71(±1.30) | 78.69(±2.26) | 69.82(±0.36) | 60.89(±1.67) | 0.286(±1.458) |
| **Context Prediction** | 93.82(±2.35) | 88.61(±1.69) | 79.10(±1.83) | 70.18(±0.50) | 62.01(±2.06) | 0.656(±1.686) |
| **Supervised** | 93.91(±2.17) | 87.56(±1.86) | 81.06(±1.96) | 71.49(±0.75) | 63.07(±0.64) | 1.330(±1.476) |
| **Masking Node + Supervised** | 93.93(±1.50) | 87.62(±1.65) | 80.43(±1.84) | 71.48(±0.86) | 62.93(±0.77) | 1.190(±1.324) |
| **Context Prediction + Supervised** | 92.80(±2.67) | 86.87(±1.89) | 80.62(±1.41) | 71.69(±0.66) | 63.99(±0.52) | 1.106(±1.430) |

Table 13: **GraphSAGE** GNN architecture on Scaffold Split. The average AUC without pretraining on all 5 datasets is 72.2%.

|  | BBBP | BACE | TOX21 | TOXCAST | SIDER | AVE GAIN |
|---|---|---|---|---|---|---|
| **No pretrain** | 74.59(±1.02) | 81.06(±0.15) | 75.72(±0.34) | 66.44(±0.27) | 63.31(±0.57) | 0(±0.470) |
| **Node Prediction** | 75.18(±0.52) | 82.98(±0.62) | 75.50(±0.33) | 67.32(±0.21) | 64.26(±0.06) | 0.824(±0.348) |
| **Context Prediction** | 74.89(±0.47) | 82.19(±0.59) | 75.45(±0.24) | 67.14(±0.02) | 64.22(±0.06) | 0.554(±0.276) |
| **Supervised** | 74.58(±0.44) | 81.56(±0.77) | 76.65(±0.22) | 68.26(±0.17) | 63.50(±0.25) | 0.686(±0.370) |
| **Masking Node + Supervised** | 75.66(±0.45) | 84.14(±0.37) | 76.92(±0.08) | 67.96(±0.20) | 64.70(±0.12) | 1.652(±0.244) |
| **Context Prediction + Supervised** | 76.41(±0.19) | 81.59(±0.49) | 77.61(±0.17) | 67.92(±0.19) | 64.92(±0.15) | 1.466(±0.238) |

## A.2 Graph Parameters: GNN layers

We also investigate the effect of the number of GNN layers in Graph pretraining. We use the recommended graph parameters in [12] (number of GNN layer=5), which are selected for best performance. Table 14 below is using node mask pretraining for GNN layer = 7. Results of GNN layer = 5 is presented in Table 2 and 4. Compared with layer 5 vs 7, the conclusion that "pretraining does not help statistically significantly" does not change.

Table 14: **Tune number of GNN layers**: Self-supervised + Rich feature. Layer number=7

| GNN layers | Methods | BBBP | BACE | TOX21 | TOXCAST | SIDER |
|---|---|---|---|---|---|---|
| **Balanced Scaffold Split** | **No pretrain** | 92.53(±1.96) | 87.01(±1.28) | 78.97(±1.78) | 69.07(±0.45) | 61.35(±1.51) |
|  | **Node Prediction** | 92.68(±2.33) | 86.98(±2.71) | 79.20(±1.97) | 69.83(±0.73) | 61.59(±1.05) |
| **Scaffold Split** | **No pretrain** | 74.64(±1.28) | 79.85(±0.02) | 75.75(±0.79) | 66.09(±0.29) | 61.81(±0.41) |
|  | **Node Prediction** | 74.94(±1.01) | 81.33(±1.12) | 75.81(±0.46) | 66.39(±0.23) | 63.80(±0.18) |

## A.3 Selection of Dataset

As many pretraining objectives to be evaluated are following [12, 22], we used the intersection dataset of the two papers, except we eliminate CLINTOX. Because CLINTOX has a significant high variance in balanced scaffold split even without pretraining (see Table 15 and 16 below), so we remove it from the evaluation to avoid unstable and biased evaluation. Additionally, we provide CLINTOX results (Table 15 and 16). The conclusion is the same on CLINTOX as on the other datasets.

Table 15: **Clintox**:High variance on **Balanced Scaffold Split**.

| No Pretrain | Layer = 5 | Layer = 7 |
|---|---|---|
| Balanced Scaffold Split | 77.47($\pm$10.44) | 77.45($\pm$9.26) |
| Scaffold Split | 89.70($\pm$0.93) | 88.50($\pm$0.85) |

Table 16: Self-supervised + rich feature + GNN layer = 7

| | Methods | BBBP | BACE | CLINTOX | TOX21 | TOXCAST | SIDER |
|---|---|---|---|---|---|---|---|
| **Balanced Scaffold Split** | No pretrain | 92.53(+/-1.96) | 87.01(+/-1.28) | 77.45(+/-9.26) | 78.97(+/-1.78) | 69.07(+/-0.45) | 61.35(+/-1.51) |
| | Node prediction | 92.68(+/-2.33) | 86.98(+/-2.71) | 79.63(+/-9.21) | 79.2(+/-1.97) | 69.83(+/-0.73) | 61.59(+/-1.05) |
| **Scaffold Split** | No pretrain | 74.64(+/-1.28) | 79.85(+/-0.02) | 88.5(+/-0.85) | 75.75(+/-0.79) | 66.09(+/-0.29) | 61.81(+/-0.41) |
| | Node predicion | 74.94(+/-1.01) | 81.33(+/-1.12) | 87.16(+/-0.99) | 75.81(+/-0.46) | 66.39(+/-0.23) | 63.8(+/-0.18) |

## A.4 Reproduce existing results

We first reprudce the results from Hu et al. [12] in Table 17, where we make sure the code base can obtain the similar results where the pretraining seems to be helpful. However as in our analysis, The three factors (feature engineering, data splits, tuning of hyper-parameters) all contribute positively to the results, and the effect of these factors could be even larger than the pretraining itself.

Table 18 shows the results we reproduced for GROVER [22]. We compared the fine-tuning results with the model initialized from the pretrained ones provided in their website. However we didn't see significant gains over the model trained from scratch. We believe the hyper-parameters matters more in this case.

## A.5 Molecular benchmark description

- **BBBP**:
  The Blood-brain barrier penetration (BBBP) dataset is extracted from a study on the modeling and prediction of the barrier permeability. As a membrane separating circulating blood and brain extracellular fluid, the blood-brain barrier blocks most drugs, hormones and neurotransmitters. Thus penetration of the barrier forms a long-standing issue in development of drugs targeting central nervous system. This dataset includes binary labels for over 2000 compounds on their permeability properties. References: Martins, Ines Filipa, et al. "A Bayesian approach to in silico blood-brain barrier penetration modeling." Journal of chemical information and modeling 52.6 (2012): 1686-1697.

- **BACE**
  The BACE dataset provides quantitative (IC50) and qualitative (binary label) binding results for a set of inhibitors of human $\beta$-secretase 1 (BACE-1). All data are experimental values reported in scientific literature over the past decade, some with detailed crystal structures available. A collection of 1522 compounds with their 2D structures and properties are provided.

  References: Subramanian, Govindan, et al. "Computational modeling of $\beta$-secretase 1 (BACE-1) inhibitors using ligand based approaches." Journal of chemical information and modeling 56.10 (2016): 1936-1949.

- **TOX21**
  The "Toxicology in the 21st Century" (Tox21) initiative created a public database measuring toxicity of compounds, which has been used in the 2014 Tox21 Data Challenge. This dataset contains qualitative toxicity measurements for 8k compounds on 12 different targets, including nuclear receptors and stress response pathways.

  The data file contains a csv table, in which columns below are used: "smiles" - SMILES representation of the molecular structure "NR-XXX" - Nuclear receptor signaling bioassays results "SR-XXX" - Stress response bioassays results please refer to the links at `https://tripod.nih.gov/tox21/challenge/data.jsp` for details.

  References: Tox21 Challenge. `https://tripod.nih.gov/tox21/challenge/`

- **TOXCAST**
  ToxCast is an extended data collection from the same initiative as Tox21, providing toxicology data for a large library of compounds based on in vitro high-throughput screening. The processed collection includes qualitative results of over 600 experiments on 8k compounds.

Table 17: Test ROC-AUC (%) performance on molecular property benchmarks using *unsupervised* and *supervised* pre-training objectives (self-supervised). Unlike the basic framework, the results are generated using *Basic* features (not rich feature), scaffold split without averaging over three different data splits, and no selection of the best performance on six learning rates. All these factors lead to conclusion in favor of pretraining.

| ICLR code | BBBP | BACE | TOX21 | TOXCAST | SIDER |
|---|---|---|---|---|---|
| **No pretrain** | 89.82 | 79.46 | 77.39 | 67.91 | 58.38 |
| **Node Prediction** | 90.42 | 84.03 | 78.70 | 68.36 | 59.97 |
| **Context Prediction** | 91.07 | 83.41 | 78.84 | 68.57 | 60.78 |
| **Supervised** | 89.37 | 82.74 | 79.78 | 68.39 | 63.30 |
| **Masking Node + Supervised** | 91.53 | 83.94 | 81.34 | 70.93 | 62.46 |
| **Context Prediction + Supervised** | 91.00 | 82.90 | 81.23 | 71.34 | 62.72 |

Table 18: Reproduce Grover [22]

| GROVER | Number of parameters | BBBP | BACE | TOX21 | TOXCAST | SIDER |
|---|---|---|---|---|---|---|
| No pretrain (base) | 48,790,038 | 92.6 (+/- 2.0) | 86.2 (+/- 2.4) | 79.3 (+/- 2.6) | 71.5 (+/- 0.3) | 62.3 (+/- 0.3) |
| No pretrain (large) | 107,714,488 | 91.8 (+/- 2.9) | 86.4 (+/- 2.1) | 79.8 (+/- 2.2) | 71.5 (+/- 4.0) | 63.6 (+/- 0.7) |
| grover pretraining (base) | 48,790,038 | 89.6 (+/- 0.8) | 85.1 (+/- 1.1) | 78.7 (+/- 2.4) | 71.5 (+/- 0.5) | 62.0 (+/- 2.4) |
| grover pretraining (large) | 107,714,488 | 89.2 (+/- 0.7) | 84.6 (+/- 1.4) | 78.6 (+/- 2.6) | 69.1 (+/- 2.4) | 61.4 (+/- 1.9) |

The data file contains a csv table, in which columns below are used: "smiles" - SMILES representation of the molecular structure "ACEA_T47D_80hr_Negative",
"Tanguay_ZF_120hpf_YSE_up" - Bioassays results please refer to the section "high-throughput assay information" at https://www.epa.gov/chemical-research/toxicity-forecaster-toxcasttm-data for details.

References: Richard, Ann M., et al. "ToxCast chemical landscape: paving the road to 21st century toxicology." Chemical research in toxicology 29.8 (2016): 1225-1251.

- **SIDER**

  The Side Effect Resource (SIDER) is a database of marketed drugs and adverse drug reactions (ADR). The version of the SIDER dataset in DeepChem has grouped drug side effects into 27 system organ classes following MedDRA classifications measured for 1427 approved drugs.

  The data file contains a csv table, in which columns below are used: "smiles" - SMILES representation of the molecular structure "Hepatobiliary disorders"  "Injury, poisoning and procedural complications" - Recorded side effects for the drug Please refer to http://sideeffects.embl.de/se/?page=98 for details on ADRs.

  References: Kuhn, Michael, et al. "The SIDER database of drugs and side effects." Nucleic acids research 44.D1 (2015): D1075-D1079. Altae-Tran, Han, et al. "Low data drug discovery with one-shot learning." ACS central science 3.4 (2017): 283-293. Medical Dictionary for Regulatory Activities. http://www.meddra.org/