# OpenReview forum: "Does GNN Pretraining Help Molecular Representation?"
_NeurIPS.cc/2022/Conference — NeurIPS 2022 Accept_

### Official Review · Reviewer_GwSv · 2022-06-16

**Rating:** 4
**Confidence:** 4
**Soundness:** 2 fair
**Presentation:** 4 excellent
**Contribution:** 3 good

**Summary:**

This paper benchmarks the performance of graph pretraining under different settings. In particular, the paper performs thorough ablation studies on the key components of GNN pretraining, including pretraining objectives, data splitting methods, input features, pretraining dataset scales, and GNN architectures. They find that self-supervised graph pretraining does not have statistically significant advantages over non-pretraining methods in many settings. Although improvement can be observed with additional supervised pretraining, the improvement may diminish with richer features or more balanced data splits. The authors hypothesize the complexity of pretraining on molecules is insufficient, leading to less transferable knowledge for downstream tasks.

**Questions:**

1. Why the five downstream datasets are used? Why others are not used?

2. What the reason behind the success and failure of graph pretraining?

**Limitations:**

Yes.

**Strengths And Weaknesses:**

Strength:
1. The idea of benchmarking graph pretraining under different settings is original and deserves a thorough study.
2. This paper considers many different settings including pretraining objectives, data splitting methods, input features, pretraining dataset scales, and GNN architectures. The settings are extensive and well studied in the experiments.
3. The organization and writing are sound and satisfying.

Weakness:
1. The authors use ZINC/SAVI for self-supervised learning, ChEMBL for supervised pretraining, and evaluate the performance on BBBP, BACE, TOX21, TOXCAST, and SIDER. I am not sure whether the five datasets BBBP, BACE, TOX21, TOXCAST, and SIDER are representative enough. It seems that the choice of downstream tasks is a bit ad-hoc. For example, these three datasets ClinTox, MUV, and HIV are also widely used for evaluating graph pre-training. As far as I know, the performance of graph pretraining varies on different datasets. The results on some datasets are insignificant and may be worse than random initialization. The authors may need to provide more evidence about why these five datasets are chosen and why other datasets are not considered. Otherwise, the evaluation is not comprehensive enough.
2. The authors show that graph pretraining does not bring significant improvements in many settings. Then the authors hypothesize that the complexity of pretraining on molecules is insufficient, leading to less transferable knowledge for downstream tasks. However, the paper does not provide any insights into why graph pretraining fails in many different settings. On the other hand, graph pretraining does improve the performance in some settings. Although the paper hypothesizes that the similarity between downstream tasks and pretraining tasks might play a role here, the paper does not provide significant insights into why pretraining can help in these settings. Some empirical analysis might be needed to improve the paper.

---

> ### Author Response · Authors · 2022-08-01
> **Thank you for your review!**
>
> Thank you so much for your very helpful and constructive review! With your inputs, our paper can improve significantly, so we really appreciate your time and efforts. Here are our replies.
>
> **Q1: Why did you choose the above five datasets? Why exclude ClinTox, MUV, and HIV?**
>
> Thank you for the very good question and valid concerns! We did not cherry-pick downstream datasets.  As many pretraining objectives to be evaluated are following [1][2], we used the intersection dataset of the two papers, except we eliminate CLINTOX. Because CLINTOX has a significant high variance in balanced scaffold split even without pretraining (see Table 1 below, red), so we remove it from the evaluation to avoid unstable and biased evaluation.
>
>
> **Table 1: Clintox: high variance on Balanced Scaffold Split**
> | No pretrain | Layer=5 | Layer=7 |
> |--- |--- |--- |
> | Balanced Scaffold Split | 77.47 (`+/- 10.44`) | 77.45 (`+/-9.26`) |
> | Scaffold Split | 89.70 (+/- 0.93) | 88.50 (+/-0.85) |
>
> Additionally, we have CLINTOX results. The conclusion is the same on CLINTOX as on the other datasets.
>
> **Table 2: self-supervised + rich feature + GNN layer = 7**:
>
> |                         | Methods         | BBBP           | BACE           | **`CLINTOX`**        | TOX21          | TOXCAST        | SIDER          |
> |-------------------------|-----------------|----------------|----------------|----------------|----------------|----------------|----------------|
> | Balanced Scaffold Split | No pretrain     | 92.53(+/-1.96) | 87.01(+/-1.28) | **77.45(+/-`9.26`)** | 78.97(+/-1.78) | 69.07(+/-0.45) | 61.35(+/-1.51) |
> |                         | Node prediction | 92.68(+/-2.33) | 86.98(+/-2.71) | **79.63(+/-`9.21`)** | 79.2(+/-1.97)  | 69.83(+/-0.73) | 61.59(+/-1.05) |
> | Scaffold Split          | No pretrain     | 74.64(+/-1.28) | 79.85(+/-0.02) | **88.5(+/-0.85)**  | 75.75(+/-0.79) | 66.09(+/-0.29) | 61.81(+/-0.41) |
> |                         | Node predicion  | 74.94(+/-1.01) | 81.33(+/-1.12) | **87.16(+/-0.99)** | 75.81(+/-0.46) | 66.39(+/-0.23) | 63.8(+/-0.18)  |
>
>
>
>
>
> **Q2.1 Why pretraining for graph doesn’t work for some cases, but work in other cases?**
>
> Thank you for the important question! Our experiments show self-supervised pretraining (only molecular graph, do not have any kinds of supervision) don’t help much, whereas supervised pretraining (with additional supervision labels that related with downstream task) help to some extent.
>
> So when will pretraining help? With additional supervision (labels) may help. More detailed discussions are in Sec 5 (“When pretraining helps?” and “When the gain diminishes?”)
>
> **Q2.2 What is your insight of why self-supervised pretraining doesn’t works?**
> Another great question! Our insights are in Figure 2, where the pretraining accuracy for molecular graph plateaus at a high accuracy easily (95%+ within the first few epochs), whereas for NLP the pretraining curve hardly achieves such high accuracy (e.g. plateaus around ~70%). A very high pretraining accuracy indicates the pretrain task does not have enough complexity and not much information contained in the pretraining task, therefore less valuable information can be transferred. Then why the pretrain task doesn't have much complexity? We discussed in Sec 4.6 the reason for why molecular graph tasks being trivial (compared with NLP) is because vocabulary size of molecules is small, diversity is low, structure has restricted rules, etc.
>
>
> **Reference** \
> [1] Hu et al. Strategies for Pre-training Graph Neural Networks. \
> [2] Rong et al. Self-Supervised Graph Transformer on Large-Scale Molecular Data

---

> > ### Comment · Reviewer_GwSv · 2022-08-08
> > **Thank the authors for the reponse!**
> >
> > I have another question. What's the performance with different scales of self-supervised learning data? As you enlarge the scale of pretraining data, it may help the model generalize better?

---

> > > ### Author Response · Authors · 2022-08-08
> > > **Thank you for your great follow-up question!**
> > >
> > > Another really good point! We really appreciate your follow-up question!
> > >
> > >
> > >
> > > The original motivation for our project was actually to reproduce the remarkable success on NLP when pre-training on large-scale datasets for graphs. So we performed scale-up experiments in Tables 10 and 11, which are pretrained on the SAVI dataset (containing **1 billion molecules**), which is **500 times larger** than the standard ZINC15 (containing 2 million molecules). Compared to the standard ZINC15 pretraining datasets: Balanced Scaffold Split (Table 2 vs. Table 10) or Scaffold Split (Table 4 or Table 11), we concluded that we did not observe significant gains due to scaling up (more details in Sec 4.6 Pretraining dataset).
> > >
> > > We also investigate the reason: As Fig 2 shows that the pretraining curves of SAVI is very similar with that of standard pretrain dataset ZINC15. Both curves reach high accuracy very fast (acc=95%+ within 0.1 M and 0.2 M steps). For SAVI, 0.2M steps indicate that the model can reach very high accuracy even without seeing one epoch of the entire data. So we come to the conclusion that large scale data may not help much in the current setup.

---

> > > > ### Comment · Reviewer_GwSv · 2022-08-09
> > > > **Thanks for the reply**
> > > >
> > > > Thank the authors for the reply!
> > > >
> > > > I notice that the random initialization results seem to perform much better than the results reported by other work (https://arxiv.org/pdf/2010.13902.pdf, https://openreview.net/forum?id=HJlWWJSFDH). What's the reason behind this? Is it because the setting is different or other reasons?

---

> > > > > ### Comment · Reviewer_GwSv · 2022-08-09
> > > > > **More questions**
> > > > >
> > > > > 1. The authors provide many insights and conjectures, but not many ablation studies.
> > > > >
> > > > > 2. The baseline description on GraphCL (We leverage the contrastive learning approach in [28],…) is not accurate. The graph augmentation in GraphCL is by changing the graph topology (like edge dropping and node dropping), not corrupting the input node features with Gaussian noise to generate different views.
> > > > >
> > > > > 3. Also, could the authors confirm that their method and baselines following the same evaluation settings e.g. following the same data split since this paper uses many different splits like scaffold splitting, balanced spitting etc?

---

> > > > > > ### Author Response · Authors · 2022-08-09
> > > > > > **Thank you for your great follow-up questions!**
> > > > > >
> > > > > > We sincerely thank you for the valuable time you spent on improving our paper in the busy rebuttal time! Thanks for your useful and constructive comments. With your inputs, our paper can improve significantly. Please see our replies.
> > > > > >
> > > > > > **Q1 “I notice that the random initialization results seem to perform much better than the results reported by other work [1][2]. What's the reason behind this? Is it because the setting is different or other reasons?”**
> > > > > >
> > > > > > Thanks for asking! Our “scaffold split” is exactly following [2]. We won’t say our results are much better, instead the results are compatible. Please compare Table 1 of [2] with Table 8 or Table 9 in our paper.
> > > > > >
> > > > > > Yes, you are correct, due to setting and mainly due to different train/valid/test split. For example, Balanced scaffold (e.g. Table 6) split vs scaffold split (e.g. Table 8) give very different results.
> > > > > >
> > > > > > We haven’t found the setup details of [1] in its paper, so we cannot say for sure at this moment for [1].
> > > > > >
> > > > > > More details of “scaffold split” vs “balanced scaffold split” is in Line 186-192
> > > > > >
> > > > > > **Q2 “Also, could the authors confirm that their method and baselines follow the same evaluation settings e.g. following the same data split since this paper uses many different splits like scaffold splitting, balanced spitting etc?”**
> > > > > >
> > > > > > Thank you for the great question and we understand your concerns.
> > > > > > Yes, we confirm. All methods to be compared are evaluated under the same settings (as indicated by the table title: say “Table 8: Unsupervised + Basic feature + Scaffold.” indicating the setup we use: unsupervised objective with basic feature on scaffold split).
> > > > > >
> > > > > > The reason to provide “scaffold split” and “balanced scaffold split” is to show the effect of different splits of downstream task on influencing pretraining performance.
> > > > > >
> > > > > > **Q3: The baseline description on GraphCL (We leverage the contrastive learning approach in [28],…) is not accurate. The graph augmentation in GraphCL is by changing the graph topology (like edge dropping and node dropping), not corrupting the input node features with Gaussian noise to generate different views.**
> > > > > >
> > > > > > Thank you very much for pointing this out! You are right we didn’t use GraphCL [1]. We will be explicit about this in our paper.
> > > > > >
> > > > > > We cited GraphCL [1] (L136) because we follow [1]’s contrastive loss (eq3 of [1]), as this loss is the most popular loss for contrastive learning. Also because we want to give a general definition and context of graph contrastive learning  to the readers, and [1] is a great reference. But we don’t use the specific contrastive learning methods proposed (more specifically, augmentation methods) in GraphCL. So we later explicitly reiterate on how we generate the augmentation (i.e. gaussian noise. L136 “where we corrupt the input node features with Gaussian noise to generate different views of …”). To conclude, in our paper, we will be explicitly clear that we do not use GraphCL, but only follow its contrastive loss.
> > > > > >
> > > > > > As you know, contrastive learning is a broad concept that includes data augmentation and build contrastive loss, and there are various ways to design data augmentation and contrastive loss. GraphCL is one of the contrastive learning methods, and ours is another, and many others. As we mentioned to other reviewers, we will be explicitly clear we haven’t exhausted many variants of the contrastive learning, including GraphCL. As it is not practical to exhaust all graph pretraining methods, when we design experiments, we tried to include one representative method per category of pretraining objectives, and contrastive learning is one category.
> > > > > >
> > > > > > **Q4: The authors provide many insights and conjectures, but not many ablation studies.** \
> > > > > > Thank you for the comments and we understand your points! We would say:
> > > > > >
> > > > > > For investigating how different factors (pretraining objectives, splits, feature, etc) influence the performance of graph pretraining on downstream tasks, all the tables (Table 2-13) are ablation studies, where we change one factor at a time to observe the performance change and to investigate which factors have more effects on pretraining’s performance. This is our main goal of the paper.
> > > > > >
> > > > > > For understanding why graph pretraining doesn’t significantly help as NLP, we provide our own insights and conjectures on the reason behind the results.
> > > > > >
> > > > > >
> > > > > >
> > > > > > **Reference**:  \
> > > > > > [1] Graph Contrastive Learning with Augmentations
> > > > > > https://arxiv.org/pdf/2010.13902.pdf \
> > > > > > [2] Strategies for Pre-training Graph Neural Networks
> > > > > > https://openreview.net/forum?id=HJlWWJSFDH

---

> > > > > > > ### Comment · Reviewer_GwSv · 2022-08-10
> > > > > > > **Thanks for the reponse**
> > > > > > >
> > > > > > > I will raise the rating to 4.

---

> > > > > ### Author Response · Authors · 2022-08-09
> > > > > **Thank you for your great follow-up questions!**
> > > > >
> > > > > We merged our answers together. Please see reply under "More questions" tab. Thank you!

---

### Official Review · Reviewer_gddu · 2022-06-26

**Rating:** 3
**Confidence:** 5
**Soundness:** 2 fair
**Presentation:** 3 good
**Contribution:** 2 fair

**Summary:**

In this paper, the authors perform a comprehensive empirical study on GNN pre-training for molecular representations from the perspectives of pretraining objectives, data splitting methods, input features, pretraining dataset scales, and GNN architectures. They find that self-supervised pretraining alone does not provide statistically significant improvements over non-pretrained methods on downstream tasks. Also, data splitting methods, input features, hyperparameters may be more important than the choice of pre-training strategies. Additionally, they think the insufficient complexity of pre-training on molecular graphs is the key factor for the poor performance.

**Questions:**

See the weakness above.

**Limitations:**

Yes, the authors have adequately addressed the limitations and potential negative societal impact of their work.

**Strengths And Weaknesses:**

**Strengths**

- The paper is well-presented and easy to follow.
- The experimental results are meaningful to the community.

**Weakness**
- The authors only evaluate the performance on the downstream task of molecular property prediction. However, their used MoleculeNet benchmark is potentially brittle because the performance of pre-trained models on these datasets is extremely unsteady with diverse random seeds, which may bias their conclusions. I suggest the authors to try more recent benchmarks such as Alchemy [1] for property prediction. Additionally, the learned molecular representations can also be applied in more molecule-related downstream tasks such as drug-target Interaction [2]. Above-mentioned benchmarks or tasks may help avoid the instability of MoleculeNet.
- The conclusion of existing pre-training strategy cannot work in molecular representations may not be convincing. Firstly, the authors only evaluate some naïve strategies which pre-train GNNs at the level of either entire graphs or individual nodes. As illustrated in a representative work [3], these tasks often give limited improvement and can even lead to negative transfer on many downstream tasks. Moreover, the authors show that 'Masking Node + Supervised' or 'Context Prediction + Supervised' pre-training tasks can bring notable improvements over 'no pre-training', which aligns the conclusions of [3] that combining node-level and graph-level pre-training will be conducive indeed. Additionally, the authors only try the vanilla molecular graph augmentations (adding Gaussian noise to input node features) in contrastive learning, which may alter the semantics of molecules dramatically due to their discrete nature and thus incur negative transfer. I would suggest the authors to try more advanced molecular augmentations such as automated augmentations [4], hidden feature augmentations [5] and knowledge-guided augmentations [6].
- The discussions of the reasons why pretraining does not help are neither adequate nor convincing. Moreover, how can we make the molecular pretraining more difficult?

[1] Alchemy: A Quantum Chemistry Dataset for Benchmarking AI Models

[2] Pre-training Molecular Graph Representation with 3D Geometry (ICLR 2022)

[3] Strategies for Pre-training Graph Neural Networks (ICLR 2020)

[4] Graph Contrastive Learning Automated (ICML 2021)

[5] SimGRACE: A Simple Framework for Graph Contrastive Learning without Data Augmentation (WWW 2022)

[6] MoCL: Data-driven Molecular Fingerprint via Knowledge-aware Contrastive Learning from Molecular Graph (KDD 2021)

---

> ### Author Response · Authors · 2022-08-01
> **Thank you for your review!**
>
> Thank you so much for your very insightful and constructive feedback!  With your inputs, our paper can improve significantly, so thank you again for your time and efforts. Your concerns concentrate on the selection of (1) downstream benchmarks and (2) pretraining objectives. Please see our reply.
>
> **Q1:  MoleculeNet as downstream tasks are unstable, e.g. random seeds have variance. Why choose it? How to ensure rigorous evaluation?** \
> This is a very good point! We totally agree with you that MolecueNet is not as stable as other benchmarks for other domains (e.g. MNIST or CIFAR10, etc). To ensure a rigorous evaluation, we evaluated all the experiments with multiple random seeds to reduce variance across runs and improve robustness. We also tested different splits (scaffold or balanced scaffold) to provide multiple angles of the evaluation.
>
> **Why MolecueNet?**: As you know, MoleculeNet datasets, including multiple different datasets, are to date the most widely-used and well-recognized benchmarks for molecular property of molecular graph applications, e.g. which is used in [2-8]. As a paper focused on molecular property, we want to ground our analysis with the standard norm of the community.  Regarding instability, we think it is because molecular graphs are heterogeneous and the molecular property class is imbalanced.  We think the instability is more because of the challenging application, instead of the benchmarks.
>
>
> **Q2.1 How you choose graph pretraining objectives to evaluate? Are they naïve strategies?** \
> Thank you for the great question! We consider our paper as a third party evaluation of popular well-known pretraining methods. Those methods are selected based on community-recognition (e.g. well cited, pioneer work etc). So we won’t say they are naive. We report our findings based on our experiments of those methods. These findings can agree or disagree with the original paper (e.g. [3][7]), and we reported them regardless of agreeing with previous work or not, to provide a second voice.
>
> **Q2.2 Include more methods of “molecular augmentations for contrastive learning”:**
> Another great point! We totally agree with you that more experiments on different variants of contrastive learning will be very interesting and a good addition to our paper, which can make it a more comprehensive story! However it is not practical to evaluate all existing methods in one paper. To make a balance we tried to pick one representative work per each type of pretraining, including the category of contrastive learning. We will be explicitly clear in our paper that we didn’t exhaust many variants of contrastive learning methods.
>
> Thank you for the awesome suggestions of [4][5][6] —- they are all very exciting and elegant graph works! However, when considering performance, they seem not very competitive for molecular application. Comparing Table 2 or Table 4 of our paper with Table 6 of [4]; Table 3 of [5]; Table 3 of [6], our results are better than almost all the comparison (at least 2% + for every dataset). As performance is critical for graph-pretraining, we would go with the ones with higher performance (and ideally simpler implementation).
>
> **Q3 How can we make molecules more difficult?** \
> Thanks for the very good question! Our personal opinions are: maybe we shouldn’t do that at the first place ;-) Pretraining is not the only way for graph research. After doing all these experiments, our takeaways are: if we lead a graph project for molecular property application, we will not focus our time and efforts on using pretraining to boost accuracy at the beginning; there are tens of other techniques (maybe much easier) for graph applications! So we think one should exhaust these simple techniques first before moving to pretraining.
>
> **Reference**: \
> [1] Alchemy: A Quantum Chemistry Dataset for Benchmarking AI Models \
> [2] Pre-training Molecular Graph Representation with 3D Geometry (ICLR 2022) \
> [3] Strategies for Pre-training Graph Neural Networks (ICLR 2020) \
> [4] Graph Contrastive Learning Automated (ICML 2021) \
> [5] SimGRACE: A Simple Framework for Graph Contrastive Learning without Data Augmentation (WWW 2022) \
> [6] MoCL: Data-driven Molecular Fingerprint via Knowledge-aware Contrastive Learning from Molecular Graph (KDD 2021) \
> [7] Rong et al. Self-Supervised Graph Transformer on Large-Scale Molecular Data (NeurIPS2020)

---

> > ### Comment · Reviewer_gddu · 2022-08-08
> > **Thanks for the authors' response!**
> >
> > Thanks for the authors' efforts in the response! Several vital issues remain to be addressed before the paper can be published:
> >
> > 1. As I suggested before, why not try some new tasks (such as drug-target interaction prediction) for evaluating the molecular representations? This task may be more steady than molecular property prediction using MoleculeNet.
> >
> > 2. Almost all the pre-training tasks adopted in this paper are performed at the level of either entire graphs or individual nodes, which has been validated ineffective in previous publications [1].  As a remedy, [1] performs GNNs pre-training at both levels. Hence, it is unconvincing to conclude that GNN pre-training is ineffective only adopting some pre-training tasks at the level of either entire graphs or individual nodes.
> >
> > 3. On one hand, the crucial conclusions of this empirical study have been validated in the previous study (see point 2). On the other hand, the authors did not contribute new strategies for pre-training GNNs, which hinders this work to be a strong candidate for NeurIPS main track.  I would suggest the authors polish their paper thoroughly and submit it to the NeurIPS datasets and benchmarks track for possible publication, which may be a more suitable venue for empirical studies.
> >
> > 4. The authors claim that the results in Table 2 or Table 4 of their paper are better than the reported results of JOAO [3], SimGRACE [4] and MoCL [2]. However, the experimental settings of Table 2 (balanced scaffold + rich features) or Table 4 (scaffold + rich features) are different from these works (scaffold + basic features).
> >
> > 5. A minor comment: Table 3 of MoCL [2] did not follow the transfer learning setting. I suggest the authors check the papers more carefully.
> >
> > [1] Strategies for Pre-training Graph Neural Networks (ICLR 2020)
> > [2] MoCL: Data-driven Molecular Fingerprint via Knowledge-aware Contrastive Learning from Molecular Graph (KDD 2021)
> > [3] Graph Contrastive Learning Automated (ICML 2021)
> > [4] SimGRACE: A Simple Framework for Graph Contrastive Learning without Data Augmentation (WWW 2022)

---

> > > ### Author Response · Authors · 2022-08-09
> > > **Thank you for your great follow-up questions!**
> > >
> > > We sincerely appreciate the valuable time you spent on improving our paper in the busy rebuttal time! Thank you for your clarification, and for providing informative and constructive suggestions. They are very useful.  Here are our responses.
> > >
> > > **[Re 2]: Are your pretraining tasks only graph level or node level, not combined the two levels as proposed in [1]? [1] shows pretrain on a single level is sub-optimal.** \
> > > Thank you very much for your clarification and reiteration! We actually have experiments exactly following [1]: pretrained on “both node level and graph level”. See experiments with “Masking Node + Supervised”; “Context Prediction + Supervised” in Table 3, 5, 7, 9, 10, 11, which combine the node level (predicting node attributes) with graph level (supervised tasks to predict molecular property using entire graphs) tasks, or combine the subgraph level tasks (predicting local context) with graph level (supervised tasks to predict molecular property using entire graphs) tasks.
> > >
> > > **[Re 3]: “The authors did not contribute new strategies for pre-training GNNs, which hinders this work to be a strong candidate for NeurIPS main track.”**
> > >
> > > Thank you for sharing your thoughts and suggestions! We kindly share a different opinion: we think delivering a different voice and a second thought on predominant belief that “graph pretraining helps significantly” could be as valuable to the community as proposing yet another graph-pre training method.
> > >
> > > Based on our experiments, we found simple factors (splits, features) can influence downstream tasks performance more than pretraining objectives. This hinders us to propose new pretraining objectives. Plus we investigate the reason: pretraining training curves saturate very high (0.9+) very fast (within first 0.1M steps) even when pretrained with very large datasets, indicating molecules pretraining datasets do not have sufficient complexity to bring significant gain to the downstream tasks. Therefore, we think summarizing our findings could also be useful to the community.
> > >
> > > **“[Re 4]: the experimental settings of Table 2 (balanced scaffold + rich features) or Table 4 (scaffold + rich features) are different from these works (scaffold + basic features).”** \
> > > Thank you for the great point! We actually have a “scaffold + basic features'' setting in Table 8 and Table 9. Our results are still better than most of the results of JOAO [3] and SimGRACE [4], and MoCL [2] (Yes, agreed, MoCL’s setting seems different.) Also, more important than comparing performance, if we know that simpler tricks, such as rich features and different splits, can do almost the same, or even better than graph pre-training, why not simpler?
> > >
> > >
> > > **[Re 1]: Why not use a quantum chemistry dataset Alchemy [6]?** \
> > > Thanks for the suggestion! For this paper, we are interested in molecular graphs and molecular properties and due to the limited resources we limit our conclusions in the scope of what we’ve studied, so we restrict the paper to particular molecular properties on the molecular domain, as stated in the title.
> > >
> > > **[Re 1]: Why are you not using  drug-target Interaction datasets [5]?** \
> > > Thank you for the great suggestion and we understand your concerns. We think "Drug-Target Affinity datasets" used in [5] is a good additional dataset to our work and can make the story more comprehensive, Though [5] uses MoleculeNet as its main results (Table 1 of [5]), we also need to align our work with the majority of the community.
> > >
> > >
> > > **Reference:**
> > >
> > > [1] Strategies for Pre-training Graph Neural Networks (ICLR 2020) \
> > > [2] MoCL: Data-driven Molecular Fingerprint via Knowledge-aware Contrastive Learning from Molecular Graph (KDD 2021) \
> > > [3] Graph Contrastive Learning Automated (ICML 2021) \
> > > [4] SimGRACE: A Simple Framework for Graph Contrastive Learning without Data Augmentation (WWW 2022)  \
> > > [5] Pre-training Molecular Graph Representation with 3D Geometry (ICLR 2022)  \
> > > [6] Alchemy: A Quantum Chemistry Dataset for Benchmarking AI Models

---

### Official Review · Reviewer_tcsP · 2022-07-08

**Rating:** 8
**Confidence:** 3
**Soundness:** 3 good
**Presentation:** 2 fair
**Contribution:** 3 good

**Summary:**

Self-supervised pre-training of GNNs on molecule data can be very useful for tasks in computational chemistry, e.g., drug discovery. This has motivated several works proposing pretraining methods that allegedly lead to higher downstream fine-tuning performance over non-pretrained (i.e. randomly initialized) models.

The authors of the reviewed paper challenge this claim and conduct systematic experiments revealing that the aforementioned SSL methods do not have statistically significant advantages over non-pretrained methods in many settings. Their main conclusion is that current SSL methods produce "less transferable knowledge for downstream tasks."


**Questions:**

* Are the two pre-training datasets you're using (ZINC15,SAVI) commonly used by the pre-training methods you investigate? For example, [1,2] use NCI1, and [3] uses the GEOM dataset. While the two datasets you use seem to be appropriate based on your description, one possibility leading to negligible pre-training gains might still be that they are simply not well-suited for pre-training?

[1] Graph Contrastive Learning with Augmentations, You et al, Neurips 2020.
[2] Graph Contrastive Learning Automated, You et al, ICML 2021.
[3] Evaluating Self-Supervised Learning for Molecular Graph Embeddings, Wang et al, arXiv:2206.08005.

**Limitations:**

Yes, the authors describe two relevant limitations of their work.

**Strengths And Weaknesses:**

# Strengths
The paper
* systematically investigates the sensitivity of choices in different stages of the pre-training pipeline by changing one factor at a time.
* proposes to split downstream task data not randomly like previous works but based on scaffolds. This allows for better OoD generalization evaluation since evaluating on a single split can be subject to biases.
* further examines the impact of using rich features [1], i.e., additional node features not commonly used in many previous papers.
* The paper reveals several surprising results, e.g., Table 2 shows that no pretraining method performs significantly better than non-pre-trained models.
* The paper complements other recent works reporting positive results for GNN pretraining on molecules with some negative results [1,2,3,4]. This is not to say that one must be right and the other wrong. More diversity in terms of investigated settings or experimental configurations (data split, hyper-parameters, etc.) is useful for the community overall.

# Weaknesses
Assuming that the authors conducted experiments properly and with genuine efforts to reproduce previous work's results (see my questions below), I don't see any major weaknesses in this empirical study.

Some minor writing suggestions which do not significantly impact my scores:
* tables and figures are quite misplaced; e.g., Table 3 is shown on Page 4 but eventually discussed on Page 7. Figure 2 is shown right next to section 4.7; it isn't discussed there but in section 6.
* The paper looks pretty crammed. For example, section 4.4. should be split up into multiple paragraphs.  I also see that the authors removed vertical space/margins of the section titles. I understand that this is likely due to the page limit. However, I suspect that some sentences might not add much information and can be removed to make the key messages stand out more clearly. Alternatively, maybe more details can be moved to the appendix.

[1] Self-Supervised Graph Transformer on Large-Scale Molecular Data, Rong et al, NeurIPS 2020.
[2] Graph Contrastive Learning with Augmentations, You et al, Neurips 2020.
[3] Graph Contrastive Learning Automated, You et al, ICML 2021.
[4] Evaluating Self-Supervised Learning for Molecular Graph Embeddings, Wang et al, arXiv:2206.08005.

---

> ### Author Response · Authors · 2022-08-01
> **Thank you for your review!**
>
> Thank you so much for the very insightful and helpful feedback! We really appreciate your valuable inputs! Thank you for helping to improve our paper!
>
> **Re: Better organizing the paper.**  \
> Thank you for this valuable feedback! We totally agree that text and figure should be aligned well to enable better reading experience. We should move unnecessary details to the appendix to make the main point clear.
>
> **Re: Pretraining dataset.** \
> Thank you for recommending other pretraining datasets. We following [1][2] to use pretraining tasks ZINC15 (self-supervised pretraining) and Chembl (supervised pretraining), which are quite common for molecular graph pretraining. SAVI is a much larger dataset we found to test scale-up of pretraining, which to our best knowledge, is the first one to use for pretraining. The two datasets lead to similar conclusions.
>
> **Reference:** \
> [1] Hu et al. Strategies for Pre-training Graph Neural Networks. ICLR \
> [2] Rong et al. Self-Supervised Graph Transformer on Large-Scale Molecular Data. NeurIPS 2020.

---

> > ### Comment · Reviewer_tcsP · 2022-08-07
> > **Thank you for your response!**
> >
> > Thank you for your response!

---

### Official Review · Reviewer_LNeY · 2022-07-11

**Rating:** 7
**Confidence:** 4
**Soundness:** 4 excellent
**Presentation:** 3 good
**Contribution:** 4 excellent

**Summary:**

This paper presents a comprehensive empirical study on pretraining GNNs for downstream molecule property prediction tasks, investigating aspects including features, GNN architectures, pretraining datasets, pretraining objectives, downstream task data split, and fine-tuning hyperparameters etc. It shows that self-supervised pretraining alone generally bring non-significant improvement in performance alone, while supervised pretraining helps when downstream tasks do not have richer features or more balanced data splits.

**Questions:**

It would be beneficial to also include
1) another contrastive learning baseline with node dropping and subgraph augmentation
2) the performance when using several or all the self-supervised pretraining objectives
3) some more scalable architectures.

**Limitations:**

The limitations are adequately addressed.

**Strengths And Weaknesses:**

#Strengths
- The empirical study is novel, systematic and comprehensive, probing a range of design choices made by previous works.
- The central claims, especially the set of observations in section 5, are valuable summaries for future works and well supported by the experiments.
- The paper is clearly written and easy to follow

#Weaknesses
- The specific contrastive method evaluated is not the known best fit. As shown in GraphCL(https://arxiv.org/pdf/2010.13902.pdf), the choice of data augmentation is crucial. Specifically, this work shows that the suitable augmentations for biochemical molecules should be node dropping and subgraph, instead of attribute masking used in this empirical study.
- The experiments do not consider combined self-supervision objectives.
- The paper does not include more scalable architectures (e.g. deep GNN and various transformer-based GNNs)

---

> ### Author Response · Authors · 2022-08-01
> **Thank you for your review!**
>
> Thank you so much for the very helpful and useful comments! We really appreciate your contribution to help improve our paper! Please see our response below.
>
> **Re: Contrastive pretraining variants**. \
> We totally agree with you that more experiments on different contrastive variants for pretraining are quite interesting and will help our paper to provide a more comprehensive study. Some candidates like GraphCL can be a great fit! They are top on our to-investigate list. Though due to page limit, in this paper, we did not explore this direction thoroughly and we tried to include one or two representatives for each category for comparison. We will be explicitly clear in our paper that we didn’t exhaust many variants of contrastive learning methods.
>
>
> **Re: Combined self-supervised objectives**. \
> Combined self-supervised objectives is a fantastic idea! Definitely worth future efforts. Just our focus in the paper is to evaluate existing popular methods, so we didn’t experiment with new methods.  Though we evaluated a related idea: combining self-supervised with supervised (with labels) objectives in a sequential manner.  The conclusion is supervised help a bit.
>
> **Re: Deep graph and transformer based graph**: \
> We absolutely agree with you on including a transformer based graph that will make the paper more comprehensive. Really good suggestion. Although one of reference papers we use is a graph-transformer-based method [1], we didn’t explicitly and exhaustively explore transformer-graph architecture in particular (as there are a couple of different graph-transformer methods)
>
> For deeper graphs, we actually experimented graph depth w.r.t pretraining behavior. Table 1 below is a node prediction task for layer 7. Compared with Layer 5 used in Table 2 and 4 in the paper, we observed a similar conclusion.
>
> **Table 1: Tune number of GNN layers: Self-supervised + Rich feature. `Layer number=7`**
>
>
> | GNN Layers              | Methods         | BBBP           | BACE           | TOX21          | TOXCAST        | SIDER          |
> |-------------------------|-----------------|----------------|----------------|----------------|----------------|----------------|
> | Balanced Scaffold Split          | No pretrain     | 92.53(+/-1.96) | 87.01(+/-1.28) | 78.97(+/-1.78) | 69.07(+/-0.45) | 61.35(+/-1.51) |
> |                         | Node Prediction | 92.68(+/-2.33) | 86.98(+/-2.71) | 79.2(+/-1.97)  | 69.83(+/-0.73) | 61.59(+/-1.05) |
> | Scaffold Split | No pretrain     | 74.64(+/-1.28) | 79.85(+/-0.02) | 75.75(+/-0.79) | 66.09(+/-0.29) | 61.81(+/-0.41) |
> |                         | Node Prediction | 74.94(+/-1.01) | 81.33(+/-1.12) | 75.81(+/-0.46) | 66.39(+/-0.23) | 63.8(+/-0.18)  |
>
> **Reference**: \
> [1] Rong et al. Self-Supervised Graph Transformer on Large-Scale Molecular Data. NeurIPS 2020. \
> [2] Hu et al. Strategies for Pre-training Graph Neural Networks. ICLR

---

> > ### Comment · Reviewer_LNeY · 2022-08-09
> > **Thank you for the response**
> >
> > Thank you for the response and I maintain the recommendation of accept.

---

### Official Review · Reviewer_qMss · 2022-07-12

**Rating:** 6
**Confidence:** 4
**Soundness:** 3 good
**Presentation:** 3 good
**Contribution:** 3 good

**Summary:**

The paper looks into the effect of self-supervised GNN pre-training in graphs used for molecular representation. The paper assesses the performance of various popular graph pre-training objectives and finds that the benefit is negligible in many cases. The paper also finds that when improvement is observed with supervised pre-training, the improvement may diminish with richer features or more balanced data splits. Lastly, the paper shows that hyper-parameters can have a larger impact on accuracy of downstream tasks than the pretraining tasks. Paper offers examples of when pre-training might and might not be useful, based on experimental findings. Takeaways in this paper are contradictory to some related work in the field of GNNs which claims that, similarly to NLP models, GNNs might benefit from self-supervised pre-training.

**Questions:**

- Have the authors thought about whether or not findings in this paper would translate to GNNs pretrained on graphs coming from different distributions - e.g. more general domain?

- Minor Miscellaneous Edits
- Line 18: extra space (GNNs) ,
- Line 21: typo: nurturing?
- Line 184: typo:  scenarios



**Limitations:**

Authors adequate address limitations in their paper under ‘Limitations of current study’



**Strengths And Weaknesses:**

- The research question is interesting: does pretraining help in GNNs?
- Paper is sound & authors design and perform an extensive number of experiments to support their claims
- The main question of assessing the value of pretraining is conceptually significant. The findings that pretraining might have negligible gains is interesting. However, one of the takeaways of the paper is that benefit of pretraining diminishes with richer/more high quality features. I believe this is to be expected - particularly on small datasets, as the ones used in this paper (authors also mention this under Limitations). If you have a small graph and rich enough features, then there probably isn’t much additional information to get from pretraining. Moreover, one of the benefits of GNNs is being able to learn complex representations from the graph structure itself, rather than having to use rich features. For small enough graphs, there probably isn’t that much representational information to learn from. Authors acknowledge this under Section 4.6, but I think it should be made clear that the findings in the paper apply to small graphs

---

> ### Author Response · Authors · 2022-08-01
> **Thank you for your review!**
>
> Thank you so much for your very constructive and useful comments! We really appreciate your great inputs and thank you for helping improve our paper!
>
> **Re: Be clear on “small graphs”**. \
> We totally agree with you that we should be explicitly clear that our conclusion is based on all the experiments that were done on small graphs, and more specifically, small molecular graphs. Indeed, we limit our conclusion within this domain as stated in the title.
>
> **Re: Generalization of the conclusion on different distribution of the graph:** \
> Very good question!
>
> [1] For molecular graphs \
> As graphs in molecular application are heterogeneous, the “scaffold split” is designed to test generalization ability of GNN on graphs with different distribution, by separating train/valid/test with different structured graphs. So the generalization within molecular application has already been tested.
>
> [2] For very different graphs \
> For generalization of our conclusion on different applications, for example, is graph pretraining helpful enough for very big graph, such as social networks with millions of nodes and edges? We still think more experiments are needed to make a convincing conclusion.
>
> **Re: fix**:  We also appreciate your kindly correctness of the typos. We will fix them in our paper.

---

> > ### Comment · Reviewer_qMss · 2022-08-08
> > **Thank you for your response!**
> >
> > Thank you to the authors for their response!

---

### Author Response · Authors · 2022-08-01
**Thank you for the insightful reviews!**

Thank you to all the reviewers for your very insightful comments and constructive reviews! We really appreciate your valuable time and efforts spent on improving our paper. We are glad to see all the reviewers recognize the value of our paper (though the paper is not perfect) and agree that the empirical assessment of graph pretraining is conceptually important to the community. Please see our reply to each reviewers individually below.

---

### Comment · Area_Chair_7onv · 2022-08-08
**Discussion with Authors**

Dear Reviewers! Thank you so much for your time on this paper so far.

The authors have written a detailed response to your concerns. How does this change your review?

Please engage with the authors in the way that you would like reviewers to engage your submitted papers: critically and open to changing your mind. Thank you Reviewers tcsP, gddu, and GwSv for your initial engagement!

Looking forward to the discussion!

---

### Meta-Review · Area_Chair_7onv · 2022-08-26

**Recommendation:** Accept
**Confidence:** Less certain

**Metareview:**

The reviewers were split about this paper: on one hand they appreciated the sensitivity analyses and surprising findings, on the other they were concerned about the overall contribution of the work. After going through it and the discussion I have decided to vote to accept given the clear and convincing author response. I urge the authors to take all of the reviewers changes into account (if not already done so). Once done this paper will be a nice addition to the conference!

**Award:**

No

---

### Decision · Program_Chairs · 2022-09-14

Accept